# Psychological Distress in the Republic of Serbia, the Association of Social Characteristics and Substance Use on a National Representative Sample of Serbia

**DOI:** 10.3390/ijerph20075321

**Published:** 2023-03-30

**Authors:** Milica Tadic, Zorica Terzic-Supic, Jovana Todorovic, Biljana Kilibarda, Milena Santric-Milicevic, Marija Dusanovic-Pjevic, Srboljub Milicevic

**Affiliations:** 1Clinic of Neurosurgery, Gamma Knife, University Clinical Centre of Serbia, 11000 Belgrade, Serbia; 2Faculty of Medicine, Institute of Social Medicine, University of Belgrade, 11000 Belgrade, Serbia; 3Institute of Public Health of Serbia “Dr Milan Jovanovic Batut”, 11000 Belgrade, Serbia; 4Faculty of Medicine, Institute of Human Genetics, University of Belgrade, 11000 Belgrade, Serbia; 5Clinic for Gynecology and Obstetrics, University Clinical Centre of Serbia, Faculty of Medicine, University of Belgrade, 11000 Belgrade, Serbia

**Keywords:** psychological distress, social characteristics, substance use, Serbia

## Abstract

This study examined the association between social characteristics, substance use, and psychological distress in a national representative sample of adults in Serbia. It was a secondary analysis of the National Survey on Lifestyles in Serbia: Substance Abuse and Gambling 2018. The study included a total of 2000 participants aged 18 to 65 from the general population in Serbia. Psychological distress was examined using the Kessler 6 questionnaire. There were a total of 945 male participants (47.3%) and 1055 (52.8%) female participants. The average age was 37.83 ± 13.61 years. The prevalence of a high risk of psychological distress was 5.2% (103/2000), while the prevalence of moderate risk of psychological distress was 15.2% (303/2000). Multivariate logistic regression analysis showed that being male, having poor self-rated health, having poor subjective financial status, binge drinking in the past year, and lifetime use of any illicit drug were associated with a higher likelihood of having a high risk of psychological distress. One in six adults in Serbia has a high risk of psychological distress, while one in twenty has a moderate risk. The findings of this study urge targeted actions to protect and improve the health of people in psychological distress and drug and alcohol users.

## 1. Introduction

Psychological distress refers to the symptoms of strain on mental health and includes anxiety, depressive symptoms, and stress [1].

Previous studies have found that demographic, social, and personal factors such as age, gender, marital status, and socioeconomic status have been associated with psychological distress [1,2,3,4]. Psychological distress has been negatively associated with age, but it increases among the elderly, which has been associated with less social support [2]. Among individuals with higher education, higher income, and married individuals, the risk of psychological distress is lower [2].

Women have a higher likelihood of psychological distress than men. Some authors explain these differences through biological, psychological, and social risk factors [5,6,7]. Biological factors involve, e.g., hormonal changes [6], while social risk factors include the role in society and expectations for women and men in a business and family environment as well as the challenges in combining them [5].

Marriage has been found to be associated with better mental well-being in both sexes compared to widowhood and divorce. Becoming widowed has more far-reaching consequences among men than among women [5]. Financial problems are a risk factor for mental disorders [5,8]. It is not just poverty that causes psychological distress but also the stigma associated with receiving public assistance [5,9].

Various studies have found an association between smoking, substance use, and mental health [10,11,12]. Lifestyle factors, such as smoking, heavy alcohol intake, and drug use, are risk factors associated with an increased risk for psychological distress [11,12]. The strength of these associations depends on the substance type, the pattern of their use, and sociodemographic characteristics [10]. Another factor that was shown to mediate the relationship between substance use and psychological distress is the age at which the first use of substances was initiated [13].

It is essential to understand the factors that lead to psychological distress and affect the population, especially the working-age population, to avoid mental, physical, and emotional difficulties and exhaustion associated with illness and inability to work. The inability to work among the working-age population has multiplied individual consequences and numerous social, familial, and economic consequences related to the absence from work. Previous periods of uncertainty and challenges affect not only the person but also society. This is especially evident in times of pandemics or other social unrest, and it could lead to an increased need for interventions for preventing mental health problems, especially for high-risk groups [14]. Our study could help policymakers to implement effective countermeasures and prioritize interventions for high-risk populations.

The aim of this study was to examine the association between social characteristics, substance use, and psychological distress in the national representative sample of adults in Serbia.

## 2. Materials and Methods

The study was a secondary analysis of the data from the National Survey on Lifestyles of Citizens in Serbia: Substance Abuse and Gambling 2018 [15]. The National Survey on Lifestyles: Substance Abuse and Gambling 2018 in Serbia was a cross-sectional study that included a total of 2000 participants aged 18 to 65 and recruited from the general population in Serbia.

The quota non-probabilistic sampling was used as a method for sampling. Two categories were used to determine the stratum: the type of settlement (two categories—urban and rural) and the region (three categories—Belgrade, Vojvodina, and the rest of Serbia). The households in which the respondents were surveyed were chosen at random, with respect to defined quotas. Any member of the selected household could be interviewed if it corresponded to the quota sample plan given to the interviewer in advance [15].

Trained interviewers conducted data collection via tablet. The face-to-face computer-assisted personal interviewing (CAPI) method was applied. The questionnaire was developed for the National Survey on Lifestyles in Serbia: Substances Abuse and Gambling 2018, based on the European Model Questionnaire (EMQ) developed by the European Monitoring Centre for Drugs and Drug Addiction (EMCDDA). The questionnaire contained 158 items. The study was conducted between November and December of 2018. The response rate was 90%. Participants gave informed consent after being given a written description of the study process and aims.

Individuals excluded from the research were incarcerated individuals, patients in hospitals or therapeutic communities, homeless individuals, and individuals in elderly homes or homes for children, as well as individuals living in illegal settlements.

The study was approved by the Ethical Committee of the Republic Institute of Public Health of Serbia ‘Dr Milan Jovanovic Batut’ No. 6296/1, on 26 October 2021.

The study included a total of 13 variables. These variables were age, sex, residence, marital status, education, self-rated health, subjective financial status, anti-anxiety medications in the past 30 days, binge drinking in the past year, smoking status, alcohol consumption, any illicit drug use, and psychological distress.

Psychological distress was measured using the Kessler six-item questionnaire. Kessler’s six-item questionnaire was used to measure distress based on a question about anxiety and depressive symptoms experienced in the most recent four-week period [16]. Answers were given on a 5-point Likert scale, ranging from never (1) to always (5). Participants were classified into three categories corresponding to the score on this scale. The first category was no risk of distress (score ≤ 7 points), the second category was a moderate risk of distress (scores between 8 and 12 points), and the third category was a high risk of distress (scores of ≥13) [16].

Marital status was defined with the question ‘What is your marital status?’ (married/permanent relationship, single). We merged the categories of ‘divorced’, ‘never married’, and ‘widowed’ into ‘single’.

Self-rated health was defined with the question, ‘How would you describe your health?’ (very good, good, average, poor, very poor). We then merged the categories of ‘very good’ and ‘good’ into one ‘good’ and the categories ‘poor’ and ‘very poor’ into the category ‘poor’.

Subjective financial status was defined with the question, ‘How would you describe your financial status?’ (very good, good, average, poor, very poor). We merged the categories of ‘very good’ and ‘good’ into one ‘good’ and the categories ‘poor’ and ‘very poor’ into the category ‘poor’.

The use of anti-anxiety medications in the past 30 days was defined with the question, ‘Have you used any of the medications for calming down in the past 30 days?’ with answers yes/no.

Alcohol consumption was defined with the question, ‘How often in the past 12 months have you drunk beer, wine, spirits (e.g., vodka, gin, whiskey, cognac, brandy) or any other alcoholic drink, even in small quantities, e.g., a glass of beer, wine or spirits?’ (Every day, 5–6 times a week, 3–4 times a week, 1–2 times a week, 2–3 times a month, once a month, 6–11 times a year, 2–5 times a year, once a year, not in the past 12 months, I have not drunk in the past 12 months, but I have been drinking before and I have never been drunk in my life). Consumers were all those who declared that they had consumed alcohol at least once in the last 12 months.

Binge drinking in the past year was defined with the question, ‘How often in the past 12 months have you consumed 60 or more grams of alcohol on occasion, which is, e.g., 1.5 L of beer (e.g., three glasses/bottles/cans of 0.5 L or five glasses/bottles/cans of 0.3 L beer), OR 0.6 L wine (three glasses of 0.2 L) OR 0.18 L of spirits (six glasses of 0.03 L spirits) drinks) or any other combination?’ (Every day, 5–6 times a week, 3–4 times a week, 1–2 times a week, 2–3 times a month, once a month, 6–11 times a year, 2–5 times a year, once a year and not in the past 12 months). We merged the categories of ‘every day’, ‘5–6 times a week’, ‘3–4 times a week’, ‘1–2 times a week’, ‘2–3 times a month’, ‘Once a month’, ‘6–11 times a year’, ‘2–5 times a year’, ‘once a year’ into ‘yes’ and ‘not in the past 12 months’ into ‘no’.

The question, ‘Have you ever smoked tobacco?’ was used to assess smoking status, and participants were divided into current smokers and non-smokers. The ‘ex-smokers’ (used to smoke, not smoking for more than 12 months) were classified as ‘non-smokers’.

Any illicit drug use was assessed with the question, ‘Have you ever used cannabis (marijuana and hashish), ecstasy, amphetamines, cocaine, heroin and opiates, LSD and hallucinogenic mushrooms, new products that mimic various substances and volatile solvents (glue, thinner, gasoline, paints, varnishes, etc.)?’ with possible answers yes/no. We merged the categories of ‘cannabis (marijuana and hashish)’, ‘ecstasy’, ‘amphetamines’, ‘cocaine’, ‘heroin and opiates’, ‘LSD and hallucinogenic mushrooms’, ‘new products that mimic various substances’, ‘volatile solvents (glue, thinner, gasoline, paints, varnishes, etc.)’ into ‘Any illicit drug’.

Statistical analyses were performed using descriptive and analytical statistics. Differences between the categorical variables were examined using the chi-square test. The differences in means in numerical variables were examined using univariate ANOVA. All the variables which were shown significant were entered into two models of the multivariate logistic regression analyses. In the first model, the outcome variable was a high risk of psychological distress compared to no distress, and in the second model, the outcome variable was a moderate risk of distress compared to no distress. All statistical analyses were performed using the Statistical Package for Social Sciences (SPSS) 22.0 (Armonk, NY, USA).

## 3. Results

The study included a total of 2000 participants; 945 were male (47.3%), while 1055 (52.8%) were female; the average age of our participants was 37.83 ± 13.61 years. The prevalence of a high risk of psychological distress was 5.2% (103/2000), while the prevalence of a moderate risk of psychological distress was 15.2% (303/2000). There were significant differences between the participants with no risk of psychological distress, with a moderate risk of distress, and with a high risk of psychological distress in average age (36.73 ± 13.41 vs. 41.19 ± 13.47 vs. 44.99 ± 13.46, *p* < 0.001), sex, residence, education, self-rated health, subjective financial status, use of anti-anxiety medications in the past 30 days, binge drinking in the past year, smoking status, and lifetime illicit drug use. The characteristics of the participants are presented in Table 1.

Multivariate logistic regression analysis showed that being male (OR: 1.95, 95% CI: 1.15–3.27), having poor self-rated health (OR: 7.10, 95% CI: 3.93–12.86), having poor subjective financial status (OR: 3.70, 95% CI: 1.11–12.40), binge drinking in the past year (OR: 1.84, 95% CI: 1.04–3.26) and lifetime use of any illicit drug (OR: 2.15, 95% CI: 1.02–4.51) was associated with a higher likelihood for having a high risk of psychological distress. The results of the multivariate logistic regression analysis with a high risk of psychological distress as an outcome variable are presented in Table 2.

Multivariate logistic regression analysis showed that being male (OR: 2.10, 95% CI: 1.56–2.82), having poor (OR: 3.46, 95% CI: 2.32–5.17), or average (OR: 1.95, 95% CI: 1.37–2.78) self-rated health, having poor subjective financial status (OR: 1.95, 95% CI: 1.16–3.28), using anti-anxiety medications in the past 30 days (OR: 2.23, 95% CI: 1.53–3.26), binge drinking in the past year (OR: 2.15, 95% CI: 1.58–2.92), being a non-smoker (OR: 1.52, 95% CI: 1.12–2.05), and lifetime use of any illicit drug (OR: 1.86, 95% CI: 1.23–2.81) was associated with having a moderate risk of psychological distress. The results of the multivariate logistic regression analysis with a moderate risk of psychological distress as an outcome variable are presented in Table 3.

## 4. Discussion

The aim of this study was to examine the association between social characteristics, substance use, and risk of psychological distress in a national representative sample of adults in Serbia. In this cross-sectional study, we found that several factors are associated with the risk of psychological distress. We have shown that one in six adults in Serbia has a high risk of psychological distress, while one in twenty has a moderate risk. Our analysis showed that being male and perceiving your own financial status as bad, having poor self-rated health, binge drinking in the past year, and lifetime use of any illicit drug were associated with a higher likelihood of having a high risk of psychological distress and point towards a need for a stronger focus on these population subgroups.

Earlier studies have shown similar evidence regarding sex differences and psychological distress [5]. Viertio et al. [5] showed that psychological distress is a quite common problem where 11% of women and 8.8% of men in the nationally representative Finnish working population had psychological distress. Moreover, a large survey in the United States reported 15.1% of moderate psychological distress and 3.1% of severe distress over the 2001–2012 period [5]. The authors stated that with the different rating scales and cut-off scores used in previous research, the given prevalence figures of psychological distress are not correctly correlative between countries [5]. Matud et al. also showed that women scored higher than men in psychological distress [2].

Previous studies have likewise found that financial difficulties constitute a notable risk factor for psychological distress [9,17,18]. One previous study found that being a man with a low household income was associated with psychological distress, which is consistent with findings from our research [18]. Through different mechanisms, poor material living conditions may affect mental health, including poor social networks and restricted access to health care services [18]. When comparing the risk for experiencing common mental disorders among sexes, men and women seem different when classified by income category; in all other categories except the lowest one, women’s risk is higher than men’s risk, but financial difficulties in covering household costs seem to have equally harmful effects on mental health in both [5].

In line with previous research, binge drinking and the use of any illicit drug were associated with more psychological distress [11,12,19]. In a population-based study of adolescents, they found that alcohol use and all specific combinations of substance use were significantly associated with medium and high psychological distress. Moreover, they noted that according to their analyses, the associations between substance use and psychological distress differed across regions. In particular, substance use was not associated with psychological distress in the Eastern Mediterranean region [19]. In previous studies on the Serbian population, psychological distress was associated with smoking status [10] and problem gambling as outcome variables [20], while there was no association between psychological distress and binge drinking [21]. Additionally, sex-specific role expectations and norms, such as associating drinking alcohol with masculinity, may be related to the male preponderance of drinking [22,23]. While alcohol could be used to reduce anxiousness in some individuals, heavy alcohol consumption leads to anxiety, distress, and depression, which in turn can lead to higher levels of alcohol consumption, leaving the person caught in a dangerous circle [21]. A cross-sectional study of nationally representative samples of the United States (US) National Survey on Drug Use and Health, individuals age 12+, showed that adults with substance use disorders who smoke cigarettes experience more than twice serious psychological distress compared to those without substance use disorder who do not smoke cigarettes [24]. These results add further evidence for the co-treatment of substance use disorders and mental health problems. In some studies, psychological distress was associated with feelings of loneliness, symptoms of insomnia and consequential anxiety, and even suicidality [19,25]. However, many previously conducted studies on the association between psychological distress and substance use examined only specific populations, commonly adolescents and people with substance use disorders, or examined only the use of one specific substance, such as marijuana [13,19,26]. According to the information provided by the National Institute of Mental Health online in the Substance Use and Co-Occurring Mental Disorders section, individuals with substance abuse disorders may also have other mental health disorders. Individuals with mental health disorders may also fight substance use. While pointing out that individuals can have a substance use disorder and a mental disorder, this does not mean one caused the other. The studies so far demonstrate three probable modalities that could describe the co-existence mechanism of these problems [27]. One previous study investigated the genetic correlation, pleiotropy, and causal relationships between substance use and psychiatric disorder, suggesting that common risk factors such as specific genes and environmental factors may be risk factors that contribute to both substance abuse and mental disorders [28,29]. Many individuals with additional mental health conditions such as anxiety and depression may use substances as self-medication which can make symptoms worse over time and can lead to severe mental illness and cause substance disorders abuse [30]. Vice versa, substance use and substance use disorder can cause the appearance of other mental disorders due to changes in brain structures and functions [27,28].

The main strength of the present study is the high response rate. Additionally, our study is the first analysis to assess the association of social characteristics, substance use, and psychological distress in the adult population in Serbia on a large nationally representative sample of the working-age population. To the best of our knowledge, no study has yet been committed to the topic of the association between social factors and psychological distress either in Serbia or in the Western Balkans using data from a nationally representative sample, while most studies conducted in New Zealand, Australia, and the United States examined only distress as one of the independent variables in relation to other outcome variables [31].

The main limitation of our study is the cross-sectional design, which does not allow us to determine the direction or causality of the associations. Furthermore, data were obtained using a self-report questionnaire; therefore, we did not obtain detailed information, e.g., about mental disorders. Self-reporting bias, such as social desirability and recall bias, could affect the results. Other limitations related to individuals excluded from the research were incarcerated individuals, patients in hospitals or therapeutic communities, homeless individuals, individuals in elderly homes or homes for children, and individuals living in illegal settlements. Our findings do not apply to these individuals, and some settings may show a higher frequency of substance and alcohol use than the sample from the population used in this study as well as a higher frequency of psychological distress. A limitation might be the participants’ lack of desire to share information with the researchers. A possible limitation could be that some participants may have given socially desirable answers to some questions regarding substance abuse.

## 5. Conclusions

One in six adults in Serbia has a high risk of psychological distress, while one in twenty has a moderate risk. Our analysis showed that being male and perceiving one’s own financial status as bad, having poor self-rated health, binge drinking in the past year, and lifetime use of any illicit drug were associated with a higher likelihood of having a high risk of psychological distress and point toward a need for a stronger focus on these population subgroups. Therefore, this study contributes to reducing the gap in knowledge about the association of social characteristics, substance use, and psychological distress in the adult population. The findings of our study can help direct future research in developing and implementing an integrative program for people suffering from psychological distress and people with alcohol- and drug-related problems. More intensive cooperation is needed between the experts dealing with different aspects of substance and alcohol use and experts dealing with preventive activities.

## Figures and Tables

**Table 1 ijerph-20-05321-t001:** Characteristics of the participants.

Characteristics	Total N (%)	No Risk of Psychological Distress N (%)	Moderate Risk of Psychological Distress N (%)	High Risk of Psychological Distress N (%)	*p*-Value
Age	37.83 ± 13.61	36.73 ± 13.41	41.19 ± 13.47	44.99 ± 13.46	<0.001
Sex					
Male	945 (47.3)	801 (50.3)	111 (36.6)	33 (32.0)	
Female	1055 (52.8)	793 (49.7)	192 (63.4)	70 (68.0)	<0.001
Residence					
Urban	1151 (57.6)	899 (56.4)	194 (64.0)	58 (56.3)	
Rural	849 (42.5)	695 (43.6)	109 (36.0)	45 (43.7)	0.047
Marital status					
Married/permanent relationship	972 (48.6)	769 (48.2)	160 (52.8)	43 (41.7)	
Single	1028 (51.4)	825 (51.8)	143 (47.2)	60 (58.3)	0.125
Education					
Primary	374 (18.8)	264 (16.6)	64 (21.3)	46 (44.7)	
Secondary	1243 (62.4)	1009 (63.5)	185 (61.5)	49 (47.6)	
Tertiary	376 (18.9)	316 (19.9)	52 (17.3)	8 (7.8)	<0.001
Self-rated health					
Poor	245 (12.3)	116 (7.3)	74 (24.4)	55 (53.4)	
Average	300 (15.0)	216 (13.6)	68 (22.4)	16 (15.5)	
Good	1454 (72.7)	1261 (79.2)	161 (53.1)	32 (31.1)	<0.001
Subjective financial status					
Poor	816 (40.8)	582 (36.5)	157 (51.8)	77 (74.8)	
Average	972 (48.6)	824 (51.7)	125 (41.3)	23 (22.3)	
Good	212 (10.6)	188 (11.8)	21 (6.9)	3 (2.9)	<0.001
Anti-anxiety medications in the past 30 days					
Yes	238 (11.9)	120 (7.5)	70 (23.1)	48 (46.6)	
No	1762 (88.1)	1474 (92.5)	233 (76.9)	55 (53.4)	<0.001
Binge drinking in the past year					
Yes	613 (30.7)	471 (29.5)	114 (37.6)	28 (27.2)	
No	1387 (69.4)	1123 (70.5)	189 (62.4)	75 (72.8)	0.015
Smoking status					
Smoker	753 (37.7)	638 (40.0)	82 (27.1)	33 (32.0)	
Non-smoker	1247 (62.4)	956 (60.0)	221 (72.9)	70 (68.0)	<0.001
Alcohol consumption					
Yes	1754 (88.3)	1391 (87.9)	274 (91.0)	89 (87.3)	
No	232 (11.7)	192 (12.1)	27 (9.0)	13 (12.7)	0.278
Any illicit drug use					
Yes	185 (9.3)	126 (7.9)	45 (14.9)	14 (13.6)	
No	1815 (90.8)	1468 (92.1)	258 (85.1)	89 (86.4)	<0.001

**Table 2 ijerph-20-05321-t002:** Multivariate logistic regression analysis with a high risk of psychological distress as an outcome variable.

Characteristics	OR (95% CI)
Age	0.86 (1.00–0.98)
Sex	
Male	1.94 (1.15–3.27)
Female	1.0 reference category
Residence	
Urban	1.25 (1.77–2.03)
Rural	1.0 reference category
Education	
Primary	2.27 (0.92–5.59)
Secondary	1.30 (0.56–2.98)
Tertiary	1.0 reference category
Self-rated health	
Poor	7.10 (3.93–12.86)
Average	1.57 (0.78–3.17)
Good	1.0 reference category
Subjective financial status	
Poor	3.70 (1.11–12.40)
Average	1.49 (0.43–5.15)
Good	1.0 reference category
Anti-anxiety medications in the past 30 days	
Yes	4.97 (2.85–8.65)
No	1.0 reference category
Binge drinking in the past year	
Yes	1.84 (1.04–3.26)
No	1.0 reference category
Smoking status	
Smoker	1.09 (0.65–1.82)
Non-smoker	1.0 reference category
Any illicit drug use	
Yes	2.15 (1.02–4.51)
No	1.0 reference category

**Table 3 ijerph-20-05321-t003:** Multivariate logistic regression analysis with a moderate risk of psychological distress as an outcome variable.

Characteristics	OR (95% CI)
Age	1.01 (0.99–1.02)
Sex	
Male	2.10 (1.56–2.82)
Female	1.0 reference category
Residence	
Urban	1.31 (0.99–1.73)
Rural	1.0 reference category
Education	
Primary	1.05 (0.66–1.68)
Secondary	1.02 (0.71–1.46)
Tertiary	1.0 reference category
Self-rated health	
Poor	3.46 (2.32–5.17)
Average	1.95 (1.37–2.78)
Good	1.0 reference category
Subjective financial status	
Poor	1.95 (1.16–3.28)
Average	1.46 (0.87–2.45)
Good	1.0 reference category
Anti-anxiety medications in the past 30 days	
Yes	2.23 (1.53–3.26)
No	1.0 reference category
Binge drinking in the past year	
Yes	2.15 (1.58–2.92)
No	1.0 reference category
Smoking status	
Smoker	1.0 reference category
Non-smoker	1.52 (1.12–2.05)
Any illicit drug use	
Yes	1.86 (1.23–2.81)
No	1.0 reference category

## Data Availability

Data can be made available upon request.

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
