# Peer review of "Psychological Distress in the Republic of Serbia, the Association of Social Characteristics and Substance Use on a National Representative Sample of Serbia"

_ijerph, 2023, doi:10.3390/ijerph20075321_

Round 1

Reviewer 1 Report

The authors perfectly understand the limits of their study.  These limits are stated on PDF p. 8.   Do limited financial resources and substance abuse etc. cause psychological distress, or are they the result?  And is self-reporting an adequate way to measure psychological distress.  In addition, the paper contains no original research.  Instead it summarizes the research of others, whose categories may not be comparable.  In sum, it's a modest study that understands its own limits.  I don't think it does much to enhance our understanding of environmental causes (actually correlates) of psychological distress.  However, it is of some value to summarize the research of other studies, synthesizing the correlates of psychological distress. 

Author Response

Response to Reviewer 1 Comments

Point 1: The authors perfectly understand the limits of their study.  These limits are stated on PDF p. 8.   Do limited financial resources and substance abuse etc. cause psychological distress, or are they the result? 

Response 1: We appreciate the comments. We agree that our submission has the stated limits, which we have outlined on p. 8.:

“The main limitation of our study is the cross-sectional design, which does not allow us to determine the direction or causality of the associations. Furthermore, data were obtained using a self-report questionnaire; therefore, we did not get detailed information, e.g., about mental disorders. Self-reporting bias, such as social desirability and recall bias, could affect the results. Other limitations related to individuals excluded from the research were incarcerated individuals, patients in hospitals or therapeutic communities, homeless individuals, individuals in elderly homes or homes for children, and individuals living in illegal settlements. Our findings do not apply to these individuals, and some settings may show a higher frequency of substance use and alcohol than the sample from the population used in this study as well as the higher frequency of psychological distress. The limitation might be the participants' lack of desire to share information with the researchers. A possible limitation could be that some participants may have given socially desirable answers to some questions regarding substance abuse. “

Also on p. 8., we stated that “The main strength of the present study is the high response rate. Also, our study is the first analysis to assess the association of social characteristics, substance use, and psychological distress in the adult population in Serbia, on large nationally representative sample of working age population. To the best of our knowledge, no study has yet been committed to the topic of association between social factors and psychological distress neither in Serbia nor in Western Balkans using data from a nationally representative sample, while most studies conducted in New Zealand, Australia, and the United States examined only distress as one of the independent variables in relation to other outcomes variables [32]. ”

Therefore, we believe this study would contribute to reducing the gap in knowledge about the association of social characteristics, substance use, and psychological distress in the adult population and provide directions for future research.

Indeed, thank you for your efforts and time in providing feedback on our work.

Point 2: And is self-reporting an adequate way to measure psychological distress. 

Response 2: Thank you for this comment. Yes, we have also stated in the limitations on p. 8.: “The main limitation of our study is the cross-sectional design, which does not allow us to determine the direction or causality of the associations. Furthermore, data were obtained using a self-report questionnaire; therefore, we did not get detailed information, e.g., about mental disorders. Self-reporting bias, such as social desirability and recall bias, could affect the results. Other limitations related to individuals excluded from the research were incarcerated individuals, patients in hospitals or therapeutic communities, homeless individuals, individuals in elderly homes or homes for children, and individuals living in illegal settlements. Our findings do not apply to these individuals, and some settings may show a higher frequency of substance use and alcohol than the sample from the popula-tion used in this study as well as the higher frequency of psychological distress. The limitation might be the participants' lack of desire to share information with the researchers. A possible limitation could be that some participants may have given socially desirable answers to some questions regarding substance abuse.“

Point 3: In addition, the paper contains no original research.  Instead it summarizes the research of others, whose categories may not be comparable. 

Response 3: Thank you, we do agree that we did not actually do the field research. However, this is the secondary analysis of the original data from the study on the “National Survey on lifestyles of citizens in Serbia: Substance abuse and Gambling 2018“, so it allows us to make a more in-depth analysis compared to the previously published descriptive report of the data in Serbian population [ Kilibarda, B.; Nikolic, N. Истраживање О Стиловима Живота Становништва Србије 2018. Године, Available online: https://www.batut.org.rs/download/StiloviZivotauSrbiji2018.pdf ].

Point 4: In sum, it's a modest study that understands its own limits.  I don't think it does much to enhance our understanding of environmental causes (actually correlates) of psychological distress.  However, it is of some value to summarize the research of other studies, synthesizing the correlates of psychological distress. 

Response 4: Thank you for the kind comment. Yes, the main idea was to summarize the factors associated with psychological distress in the large, nationally representative sample.

Reviewer 2 Report

This is an interesting study looking at the relationship between social characteristics, substance abuse and psychological distress in Serbia.  I think the authors have a good command of the academic literature on the topic and are confident of the research's contribution to the academic discourse on the influences on psychological distress.  I had two main concerns with the paper as currently written.  First, in section 2 (starting at line 71), you mention a geographical sampling technique you use in the data, dividing Belgrade, Vojvodina, and the rest of Serbia.  After this mention, I did not see any discussion of the geographical differences in psychological distress in the research.  This is potentially an interesting result to the paper.  Did urbanization have an impact on psychological well being?  Did your independent variables have a different impact in different regions?  If you did not investigate the geographical variations in the data, I'm unsure why you divided up the data into three categories.  If I was writing the paper, I would either explain the differences in the three categories, or I would combine them into a single global Serbia data set.  The geographic variation would be interesting, but I don't think it is required to answer your research question.  The inclusion of this discussion in the sampling method is a bit confusing to me.  

Second, I'm curious to understand the relationship between substance abuse and psychological distress.  Does substance abuse cause psychological distress, or does increased psychological distress make an individual more likely to abuse a substance?  This is critical in a regression model, where you are assuming that the independent has a measurable influence on the dependent variable.  Have you discovered any previous research that makes this connection?  

Overall, I enjoyed the article.  I think you have a solid command of the academic literature.  Your methods are appropriate for the research question and your conclusions logically follow from your model.  This is an interesting project.  Best of luck in your future research.  

Author Response

Response to Reviewer 2 Comments

Point 1: This is an interesting study looking at the relationship between social characteristics, substance abuse and psychological distress in Serbia.  I think the authors have a good command of the academic literature on the topic and are confident of the research's contribution to the academic discourse on the influences on psychological distress. 

I had two main concerns with the paper as currently written.  First, in section 2 (starting at line 71), you mention a geographical sampling technique you use in the data, dividing Belgrade, Vojvodina, and the rest of Serbia.  After this mention, I did not see any discussion of the geographical differences in psychological distress in the research.  This is potentially an interesting result to the paper.  Did urbanization have an impact on psychological well being?  Did your independent variables have a different impact in different regions?  If you did not investigate the geographical variations in the data, I'm unsure why you divided up the data into three categories.  If I was writing the paper, I would either explain the differences in the three categories, or I would combine them into a single global Serbia data set.  The geographic variation would be interesting, but I don't think it is required to answer your research question.  The inclusion of this discussion in the sampling method is a bit confusing to me.  

Response 1: Thank you for the comment. The regions were used as a sampling frame, as it was explained in the methods section in order to obtain the sample that would be representative of the entire general working-age population in Serbia. As this was a sampling frame, we explained this in the methods section. What is important to note is that different regions have all both urban and rural settlements and populations living in urban and rural settlements, which we thought could be associated with psychological distress and was examined in our study. As the regional division in Serbia is mainly administrative, and there are no actual differences between these populations, we did not examine the regional differences. Also, this is the secondary analysis of the original data from the study on the “National Survey on lifestyles of citizens in Serbia: Substance abuse and Gambling 2018“, and we described the methodology used in this survey [ Kilibarda, B.; Nikolic, N. Истраживање О Стиловима Живота Становништва Србије 2018. Године, Available online: https://www.batut.org.rs/download/StiloviZivotauSrbiji2018.pdf ].

Point 2: Second, I'm curious to understand the relationship between substance abuse and psychological distress.  Does substance abuse cause psychological distress, or does increased psychological distress make an individual more likely to abuse a substance?  This is critical in a regression model, where you are assuming that the independent has a measurable influence on the dependent variable.  Have you discovered any previous research that makes this connection?  

Response 2: We thank the referee for these comments. Based on the obtained results, we assumed that there is an association between the examined variables, but not the way in which they are associated due to the study design that we applied in this work.

In the Discussion section, on p.7., we modestly stated the association between the research variables and also in line 220 that “In line with previous research, binge drinking and the use of any illicit drug were associated with more psychological distress [11,12,19]. In a population-based study of adolescents, they found that alcohol use and all specific combinations of substance use were significantly associated with medium and high psychological distress. Also, they noted that according to their analyses, the associations between substance use and psychological distress differed across regions. In particular, substance use was not associated with psychological distress in the Eastern Mediterranean region [19]. In previous studies in the Serbian population, psychological distress was associated with smoking status [10] and problem gambling as outcome variables [20]  while there was no association between psychological distress and binge-drinking [21]. Also, sex-specific role expectations and norms, such as associating drinking alcohol with masculinity, may be related to the male preponderance of drinking [22,23]. While alcohol could be used to reduce anxiousness in some individuals, heavy alcohol consumption leads to anxiety, distress, and depression, which in turn can lead to higher levels of alcohol consumption, leaving the person caught in a dangerous circle [21]. Cross-sectional study of nationally representative samples of the United States (US) National Survey on Drug Use and Health, individuals age 12+, have shown that adults with substance use disorders who smoke cigarettes experience more than twice serious psychological distress compared to those without substance use disorder who do not smoke cigarettes [24]. These results add further evidence for co-treatment of substance use disorders and mental health problems.”

Based on the given constructive question, we would like to make a supplement in our work in Discussion section , and add that on p.8. and in line 239

“In some studies, psychological distress was associated with feelings of loneliness, symptoms of insomnia and consequential anxiety and even suicidality [25,26]. However, many previously conducted studies on the association between psychological distress and substance use examined only specific populations, commonly adolescents and people with substance use disorders or examined only the use of one specific substance, such as marijuana [13,25,27]. According to the information provided by the National Institute of Mental Health online in the Substance Use and Co-Occurring Mental Disorders section, individuals with substance abuse disorders may also have other mental health disorders. Individuals with mental health disorders may also fight substance use. While pointing out that individuals can have a substance use disorder and a mental disorder, this does not mean one caused the other. The studies so far demonstrate three probable modalities that could describe the co-existence mechanism of these problems [28]. One previous study investigated the genetic correlation, pleiotropy, and causal relationships between substance use and psychiatric disorder, suggesting that common risk factors such as specific genes and environmental factors may be risk factors that contribute to both substance abuse and mental disorders [29,30]. Many individuals with additional mental health conditions such as anxiety and depression may use substances as self-medication and can make symptoms worse over time which can lead to severe mental illness and cause substance disorders abuse [31], and vice versa, that substance use and substance use disorder can cause the appearance of other mental disorders, due to changes in brain structures and functions [28,29].“

Point 3: Overall, I enjoyed the article.  I think you have a solid command of the academic literature.  Your methods are appropriate for the research question and your conclusions logically follow from your model.  This is an interesting project.  Best of luck in your future research

Response 3: We thank the referee for your efforts and time in providing feedback on our work. We are glad that you enjoyed the article and gave constructive feedback to improve our project, as well as guidelines for some future research.

Reviewer 3 Report

1. The authors presents the text in question very well and thematically analyzes it with the help of appropriate literature using accurate quotations and references.

2. The article is well written; the writing standard is acceptable; the topic is interestingly covered. The topic of the article is in accordance with the content, and the abstract is appropriate to the content.

3. The subject of the article is consistently and clearly presented, and the topic is very well analyzed.

4. The conclusions of the paper are clearly stated and tie together all the elements of the paper.

5. The article can be published without further changes.

6. It is certain that the reader will find the topic and the sources discussed here very interesting.

7. The paper contributes significantly to the study of public mental health situation in Serbia and its conclusions can help the targeted groups; therefore, I think it would be beneficial to print this article in "International Journal of Environmental Research and Public Health", as the topic and the content of the article is relevant and appropriate to the scientific profile of this journal.

Author Response

Response to Reviewer 3 Comments

Point 1: 1. The authors presents the text in question very well and thematically analyzes it with the help of appropriate literature using accurate quotations and references.

  1. The article is well written; the writing standard is acceptable; the topic is interestingly covered. The topic of the article is in accordance with the content, and the abstract is appropriate to the content.
  2. The subject of the article is consistently and clearly presented, and the topic is very well analyzed.
  3. The conclusions of the paper are clearly stated and tie together all the elements of the paper.
  4. The article can be published without further changes.
  5. It is certain that the reader will find the topic and the sources discussed here very interesting.
  6. The paper contributes significantly to the study of public mental health situation in Serbia and its conclusions can help the targeted groups; therefore, I think it would be beneficial to print this article in "International Journal of Environmental Research and Public Health", as the topic and the content of the article is relevant and appropriate to the scientific profile of this journal.

Response 1: We appreciate the comments. Thank you for your efforts and time in providing feedback on our work.

Round 2

Reviewer 2 Report

I would like to thank the authors for addressing my concerns with the previous manuscript.  I think this version is an improvement over the previous version.  After reading this version and the authors' response to my comments, I don't have any significant comments on the paper.  The authors have a solid command of the academic literature, a logical methodology that answers their research question.  A solid analysis of the data and conclusions that logically flow from the analysis and are well linked back to the academic literature.  I appreciate the inclusion of a discussion on the potential of psychological stress being either a cause or a result of substance abuse.  I see that as a critical aspect of their research.  Best of luck in your future research.  Thank you again for the opportunity to review your article.